# Bone Turnover Markers during Growth Hormone Therapy for Short Stature Children Born Small for Gestational Age

**DOI:** 10.3390/biomedicines12081919

**Published:** 2024-08-21

**Authors:** Alicja Korpysz, Maciej Jaworski, Ewa Skorupa, Mieczysław Szalecki, Mieczysław Walczak, Elżbieta Petriczko

**Affiliations:** 1Department of Endocrinology and Diabetology, “The Children Memorial Health” Institute, 04-736 Warsaw, Poland; m.szalecki@ipczd.pl; 2Department of Biochemistry, “The Children Memorial Health” Institute, 04-736 Warsaw, Poland; m.jaworski@ipczd.pl (M.J.); e.skorupa@ipczd.pl (E.S.); 3Department of Paediatrics, Endocrinology, Diabetology, Metabolic Diseases and Cardiology of Developmental Age, Pomeranian Medical University, 70-204 Szczecin, Poland; m.walczak@pted.pl (M.W.); elzbietapetriczko@gmail.com (E.P.)

**Keywords:** SGA, GHT, P1NP, CTX, P3NP, NT-pro-CNP, Ca-P

## Abstract

Growth hormone therapy (GHT) can improve growth velocity and final height, but can also accelerate the process of bone growth, which is related to structural bone modeling in both formation and resorption. This study evaluated the capacity of bone turnover markers to predict early growth response to one year of GHT in short stature children born small for gestational age (SGA). This study included 25 prepubertal children born SGA. We estimated P1NP (N-terminal procollagen type 1), CTX (C-terminal telopeptide of collagen type 1), P3NP (N-terminal procollagen type 3), NT-pro-CNP (amino-terminal C-type natriuretic peptide) and Ca-P metabolism using standard ECLIA (electrochemiluminescence), RIA (radioimmunoassay), and ELISA (enzyme-linked immunosorbent assay) methods. A statistically significant increase in bone resorption markers (CTX) was found at both 6 and 12 months. P1NP bone markers were increased at 6 months and after 12 months of therapy. The P3NP marker for collagen synthesis also increased after 12 months of therapy. We obtained significant increases in phosphorus levels at 6 and 12 months, and similar ALP (alkaline phosphatase) increases. We found a significant correlation between height (cm) and CTX after 6–12 months, as well as a P1NP/height (SD) correlation after 12 months. Calcium levels significantly correlated with height (SD) after 12 months. We found strong reactions of bone resorption and bone formation markers during growth hormone therapy, which may determine their selection as predictors of GHT outcome in children born SGA. However, the issue requires further research.

## 1. Introduction

Bone turnover markers represent the most common form of bone metabolism. Biochemical markers of bone remodeling may be useful in the clinical investigation of bone turnover in children [1].

Bone mineralization in children born small for gestational age (SGA) has been the subject of many studies. Van de Lagemaat showed that six months post-term, preterm infants born SGA have lower bone accretion, and in another work she proved decreased collagen type I synthesis in children born SGA [2,3]. Longhi also confirmed that children born SGA seem to have smaller and weaker bones [4]. Children born SGA had lower BMD (bone mineral density) in mid-childhood compared to children born appropriate for gestational age (AGA) in Nordman’s as well as Maruyama’s studies [5,6]. Buttazzoni’s work reports that preterm individuals born SGA are at risk of reaching low adult bone mass, partly due to a deficit in the accrual of bone mineral during growth [7]. Balasuriya confirmed that low birth weight groups displayed lower peak bone mass and a higher frequency of osteopenia/osteoporosis [8]. Serum carboxyterminal propeptide of type 1 procollagen (P1CP) and cross-linked carboxyterminal telopeptide of type 1 collagen (1CTP) were similar in infants born SGA and born AGA, but lower BMC (body mass composition) and osteocalcin were found in children born SGA [9]. The Rojo-Trejo study reports lower BMC and BMD in children born SGA than in children born AGA [10]. Finally, Silvano proved that COL1A1 polymorphism could be a useful predictor of osteopenia in children born SGA [11].

Therapy with growth hormone clearly increases markers of bone formation and resorption [12,13,14,15,16,17,18,19,20,21,22,23,24,25]. Bone turnover markers can be also an important efficacy parameter in growth hormone therapy. Previous research has confirmed the beneficial effect of growth hormone on bone turnover markers. Administration of growth hormone improves growth velocity and final height, and accelerates the process of bone growth relating to structural bone modeling, in both formation and resorption [23,24]. The Rauch study concluded that bone markers and collagen metabolism provide early indications of the therapeutic success of GH therapy in children with growth hormone deficiency (GHD), but the prediction of an individual marker is too imprecise [25]. The factors that underlie these processes are still unclear. Most studies were conducted with children with GHD, although it is rare for children with idiopathic short stature to have this condition [12,13,14,15,16,17,18,19,20,21,25,26].

The effect of growth hormone therapy (GHT) on bone turnover in children small for gestational age has been assessed in several studies [24,27,28,29,30,31,32]. P1NP (N-terminal procollagen type 1), CTX (C-terminal telopeptide of collagen type 1), P3NP (N-terminal procollagen type 3), and NT-pro-CNP (amino-terminal C-type natriuretic peptide) seem promising [24,25,26,27,28,29,30,31,32,33,34].

In this study, these markers were assessed for the first time to predict the early growth response to GHT in short stature Polish children born SGA. The novelty of this study, which examined the relationship between the effect of treatment with GHT and changes in biochemical bone metabolism markers in 25 children born SGA, is that it was conducted on Polish subjects.

We also evaluated Ca-P metabolism in these processes.

## 2. Material and Methods

### 2.1. Study Population

A total of 25 prepubertal children born small for gestational age (SGA) participated in this study (age 7.6 years ± 1.8). Children were born at 36.4 Hbd ± 4.11 with a body weight of 1776.5 g ± 699 (−3.3 SD) and a body length of 449 cm ± 6.9 (−1.2 SD). The pre-treatment height was 111.8 cm ± 10.3 (−2.7 SD), with a weight of 16.6 kg ± 3.4 (−3.34 SD) and a BMI of 13.1 ± 0.85 m^2^/kg (−2 SD) (Table 1).

Inclusion criteria were as follows: patients born SGA, a growth deficiency < −2 SD, a GH level (>10 ng/mL) in two stimulation tests, and prepubertal at the time of inclusion. Exclusion criteria were as follows: children with GHD, children born AGA (appropriate for gestational age), chromosomal abnormalities (such as Silver–Russell syndrome, Turner syndrome, Noonan syndrome, etc.), thyroid and cortisol deficiencies, diabetes, skeletal, urinary, and gastrointestinal disorders, metabolic defects, hepatic and renal failure, fractures in the last 6 months, immobilisation, and pharmacotherapy, such as glucocorticoids.

### 2.2. Methods

All parents of our patients provided written informed consent. The study protocol was approved by the Ethics Committee (6/KBE/2016). SGA was defined as birth weight and/or birth height < −2 SD for gestational age. Data on the history of prenatal and perinatal periods, as well as medical history, the curve of growth, growth velocity, parental height, and family medical history were collected. All investigations to assess short stature were performed, including two growth hormone stimulation tests with glucagon and arginine, IGF1 (somatomedin C), thyroid status, LFP (alphafetoprotein), bHCG (gonadotrophine human chronicle), transglutaminase antibody, prolactin, and cortisol concentrations, as well as blood karyotype in girls (all basic parameters: morphology, electrolytes, serum creatinine, urea, glucose, transaminases, and GGTP (gamma-glutamyltranspeptidase, bilirubin) levels). Bone age (BA), as well as head and pituitary MRIs (magnetic resonance images) were assessed. GHT was conducted at the dose 0.035 mg/kg/day.

Before treatment (baseline 0), as well as at 6 and 12 months, all anthropometric measurements were taken. Height was measured with the Harpenden stadiometer. BMI (body mass index) was expressed as height in m^2^/weight in kg. All measurements were calculated as z-score (ZS). BA, IGF1, as well as Ca, P, ALP, PTH (parathyroid hormone), 25OHD3 (25-hydroxy vitamin D3) levels, P1NP (N-terminal procollagen type 1), CTX (collagen type 1 cross-linked C-telopeptide), P3NP (procollagen 3 amino-terminal propeptide), and NTproCNP (amino-terminal C-type natriuretic peptide) were estimated at baseline, as well as at 6 and 12 months during treatment. BA was calculated by the Greulich–Pyle method. IGF1 was measured by the automatic ECLIA method on Liaison. Ca-P, ALP, and 25OHD3 levels were estimated using the automatic method in the IDS-iSYS closed immunodiagnostic system. 25-hydroxy vitamin D and PTH were measured with the immunoradiometric assay (RIA).

P1NP and CTX were estimated by the automatic ECLIA method on the Elecsys apparatus. P1NP was measured by ECLIA on the Cobas e411 analyser using Roche Diagnostics’ total P1NP assay (Roche Diagnostics, Mannheim, Germany). The total P1NP assay used monoclonal antibodies and recognised both the trimeric and monomeric forms of P1NP. The total assay time was 18 min. Stable ruthenium complexes [(Ru(bpy)_3_)^2+^] and tripropylamine (TPA) were used in the chemiluminescence reaction of the Cobas e411 system. The results were read from the calibration curve prepared on the basis of a 2-point calibration (the standard curve contained in the reagent barcode). Measuring range: 5–1200 ng/mL; lower limit of detection: 5 ng/mL. CTX measuring range: 0.010–6.00 ng/mL; lower limit of detection: 0.010 ng/mL.

P3NP assessment was conducted by radioimmunoassay (RIA) using the UniQ P3NP reagent kit by Aidian (Espoo, Finland). The UniQ P3NP kit was based on a competitive radioimmunoassay technique. The concentration of P3NP in test samples was calculated from the standard curve. The radioactivity of the samples was measured on an Elmer Wizard gamma counter. Test measurement range: 1.0–50 μg/L; detection limit, 0.4 μg.

The serum level of NT-proCNP was measured by the ELISA method (Biomedica, Makati, Philippines), by binding it with a coating antibody to form a sandwich complex, followed by the addition of a substrate (TMB, tetramethylbenzidine). Colour intensity was measured at a wavelength of 450 nm (reference wavelength 630 nm) on a multifunctional Varioscan Flash reader using SkanIt Software 2.4.3. The standard curve was developed as recommended by the reagent kit manufacturer. Based on the measurements of the standards, the reader software calculated the concentrations (pmol/L) of the control and test samples. Test measuring range: 0–128.0 pmol/L.

### 2.3. Statistical Methods

Data were analysed using Statistica PL, version 10. Results were presented as mean and standard deviation. Departures from the normality of the distribution of analysed variables were tested with the Shapiro–Wilk test. Pearson correlations were used to assess associations between variables. Adjustments were conducted with a linear regression model. Non-normally distributed variables were analysed after the Box–Cox transformation. *p*-values less than 0.05 were considered significant.

## 3. Results

The median height of our patients before treatment was 111.8 cm (−2.7 SD), and BMI was 13.1 (−2 SD). After 6 months of GHT, the mean height reached 115.82 cm (−2.46 SD) (v1–v2, *p* = 0.000000), and after one year it was 119.93 cm (−2.19 SD) (v1–v3, *p* = 0.000000) (Table 2).

IGF1 (ng/mL) levels increased significantly from the baseline to 6 (141 vs. 310, *p* = 0.000) and 12 (141 vs. 349, *p* = 0.000) months (Table 2).

Significant increases in bone resorption markers’ CTX (ng/mL) were found at both 6 months (1.7 vs. 2.4, *p* = 0.000) and 12 months (1.7 vs. 2.3, *p* = 0.000) (Figure 1). Bone markers in P1NP (ng/mL) reached a maximum at 6 months (592 vs. 840, *p* = 0.000) and after 12 months of treatment (592 vs. 804, *p* = 0.000) (Figure 2). Collagen synthesis markers P3NP (µg/mL) increased after 12 months of therapy (13 vs. 14.8, *p* = 0.000) (Figure 3 and Table 2).

In terms of Ca-P levels, we obtained significant increases in phosphorus (ng/mL) levels at 6 and 12 months (1.4 vs. 1.6, *p* = 0.000; and 1.4 vs. 1.6, *p* = 0.001, respectively), and similar increases in ALP (ng/mL) levels (231 vs. 277, *p* = 0.00013; and 231 vs. 282, *p* = 0.0012, respectively). The limit of 25OHD3 (ng/mL) levels increased after 12 months (31 vs. 35.1, *p* = 0.06). PTH (pg/mL) levels were elevated significantly after 6 and 12 months (22 vs. 24.8, *p* = 0.03) (Table 3).

### Correlations

We have found a significant correlation between height (cm) and CTX after 6–12 months (r = 0.39, *p* = 0.046 and r = 0.4, *p* = 0.046), as well as height (SD)/Ctx: r = 0.15, *p* = 0.049 after 6 months (Figure 4, Table 4). The P1NP/height (SD) correlation was important after 12 months (r = 0.44, *p* = 0.025) (Figure 5, Table 4).

Calcium levels significantly correlated with height (SD) after 12 months (r = 0.48, *p* = 0.014) (d v1–v3 Ca/height (SD) = 0.45) (Table 4).

## 4. Discussion

GH and IGF1 are anabolic peptides that increase not only growth, but also muscle mass and bone development. GH has the capacity to affect body composition and bone metabolism. For many years, the importance of GH regarding bone mass has been well documented. Bone turnover markers are markedly decreased in patients with GHD and ISS children, whereas both osteoblast and osteoclast activity increase during GH replacement therapy [12,13,14,15,16,17].

Biochemical measurements of bone turnover may be helpful in monitoring the growth process, especially during GHT. The use of bone turnover markers may be a non-invasive method in clinical practice. There are many studies concerning GH-deficient children. Kandemir et al. found that bone turnover markers (calcium, phosphate, ALP, osteocalcin, and the carboxyterminal propeptide of type 1 collagen-CPP-1) increased after 12 months of GH therapy in 39 GH-deficient children [18]. Yun Li et al. also proved an increase in bone marker carboxyterminal telopeptide of type 1 collagen (1CTP) after 6 months of growth hormone therapy in 29 children with isolated growth hormone deficiency (IGHD) and partial growth hormone deficiency (PGHT) [19]. Baroncelli et al. estimated significantly increased procollagen 1 carboxyterminal propeptide (P1CP) and 1CTP after one year of GHT in GH-deficient prepubertal children, but after 12 months P1CP declined and 1CTP levels remained stable [20]. Schweizer’s work reported early bone remodeling in prepubertal GH-deficient children treated by GH and increased P1CP [21].

A few studies estimated improvement in bone strength during GH therapy in children born SGA. The Willemsen study showed that during long-term GH treatment in short children born SGA, bone mineral density increased. The most significant results were seen in children who started treatment at a younger age and those who gained more height during GH treatment [22]. The same conclusion can be found in the study by De Zegher; the rate of bone maturation in short prepubertal children born SGA treated with GH appeared to depend not only on the dose of GH, but also on the age of the child [23]. Lem proved that during GH treatment of children born SGA, bone mineral density (BMD) increased significantly [29]. The same conclusion was reached by the Arends study, which found that GH treatment of short children born SGA increased their bone mineral density proportionately to their height gain and bone maturation [30].

In our study, we attempted to estimate changes in bone markers levels during GHT and their usefulness regarding patients born SGA treated by GH.

All patients achieved clinically significantly increased growth and IGF1 levels after one year of GHT. Our main finding shows that P1NP (bone formation marker) increased after 6 and 12 months of treatment. The mechanism related to this finding can be connected with somatomedin induced by GHT, which stimulated the proliferation and differentiation of osteoblasts that lead to activation of bone formation (which is why we conducted this study in prepubertal children in whom we could achieve uniformity of IGF1 levels). The beneficial effect of GHT may have been greater in patients with lower osteogenic activity before treatment, for whom GHT could influence the activity of growing osteoblasts. There is little research on this issue regarding children born SGA. Gascoin-Lachambre et al., in a 2007 study, evaluated thirty 7-year old children born SGA with heights < −2 SD, who were administered GHT at the same dose of 0.033 mg/kg/day. They also found increased P1NP, which was an early predictor of GH treatment, especially in ISS children, but also in children born SGA, after 6 months of therapy [24]. Our study also showed a CTX (the resorption markers) increase after 6 months and after one year of GHT in children born SGA. The increase in bone resorption may have resulted from a significant acceleration of bone metabolism due to initiating growth hormone therapy and increasing the physiological distance between bone formation and bone resorption. Lanes et al. suggested that increases in P1NP and CTX were major effects of GHT in promoting bone formation [25].

Kamp et al. assessed the GH response relationship between GH and markers of bone turnover: P1CP, P3NP, and ALP [26]; however, their study did not include children born SGA.

In the Bajoria study, the authors found a positive association between IGF1 and P1CP and a negative correlation between IGF1 and 1CTP in IUGR twins [27]. In our study, we did not find any associations between P1NP, CTX, and IGF1 levels. We found a strong correlation between CTX and P1NP markers and growth velocity.

In Shalende’s study, early serum P3NP changes were associated with GH administration, and could be a useful early predictive biomarker of an anabolic response to GH, but the study did not include children born SGA [34]. In our study, P3NP markers increased after one year of our GH therapy.

NT-proCNP was estimated as a strong marker correlated with growth velocity in children. Prickett’s study showed a direct association between NT-proCNP and the growth process. Plasma NT-proCNP was elevated at birth and fell progressively with age. The authors also emphasized the significant association between NT-proCNP and ALP levels, as well as CTX [28]. Unfortunately, we did not find any important increase in NT-proCNP or any correlation with growth velocity or IGF1.

As for Ca-P metabolism, we found a statistically significant elevation of phosphorus and ALP concentrations; Ca levels were positively correlated with growth after one year of treatment.

Bone markers are poorly documented in children, especially in patients born SGA. Even if GH and IGF1 are major factors influencing their levels, other factors are probably also involved.

This study’s limitations include the number of patients, and the length of therapy could also be a limitation of the research.

## 5. Conclusions

A strong reaction of bone resorption and bone formation markers during growth hormone therapy may determine their selection as predictors of this treatment’s outcome in children born SGA. However, the issue requires further research.

## Figures and Tables

**Figure 1 biomedicines-12-01919-f001:**
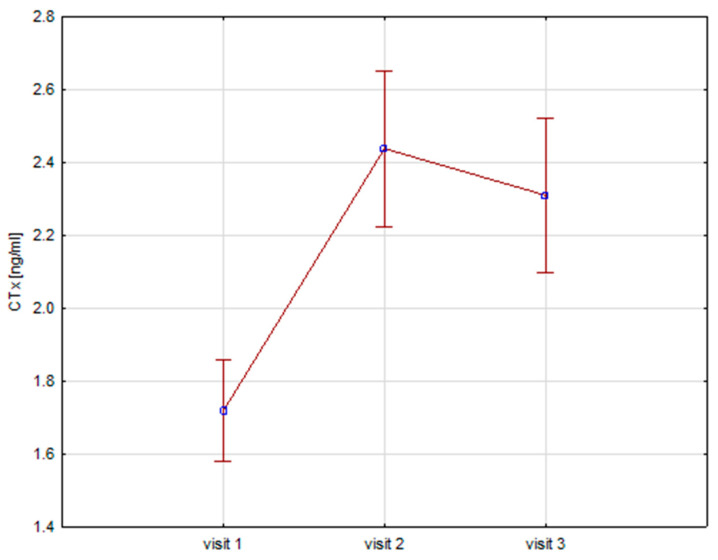
Height (v1–v3)/CTX, CTX-C-terminal telopeptide of collagen type 1.

**Figure 2 biomedicines-12-01919-f002:**
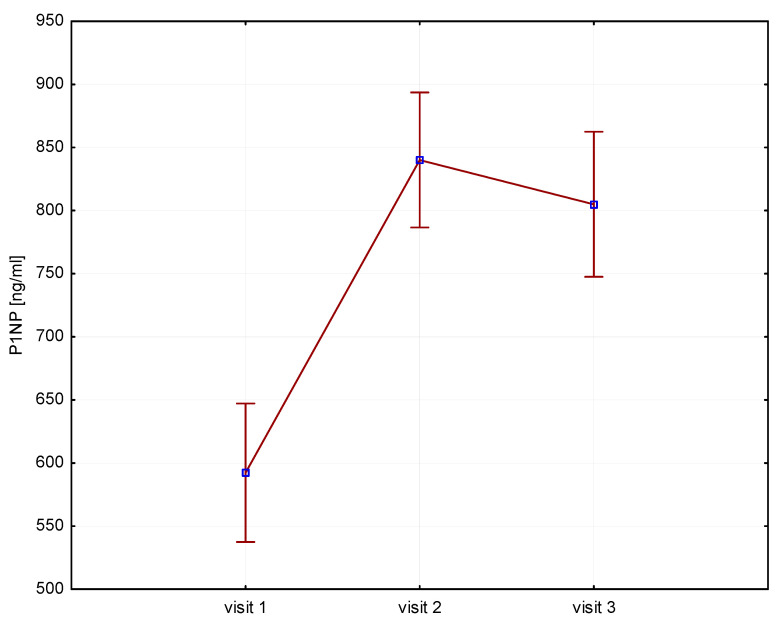
Height (v1–v3)/P1NP, P1NP-N-terminal procollagen type 1.

**Figure 3 biomedicines-12-01919-f003:**
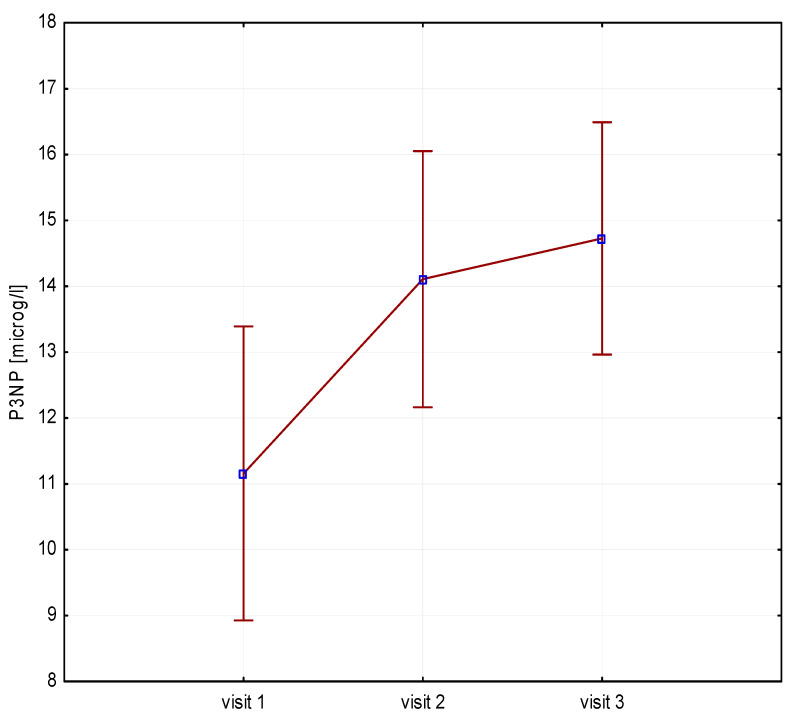
Height (v1–v3)/P3NP, P3NP-N-terminal procollagen type 3.

**Figure 4 biomedicines-12-01919-f004:**
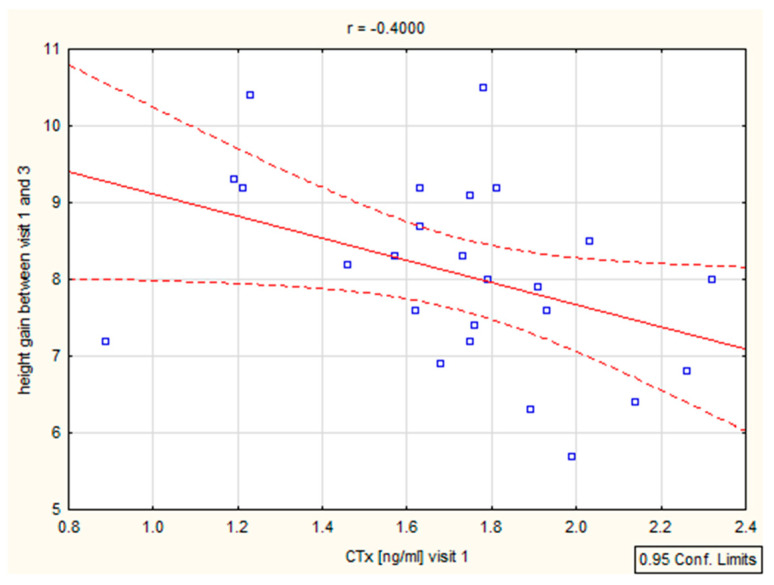
Correlation height/CTX.

**Figure 5 biomedicines-12-01919-f005:**
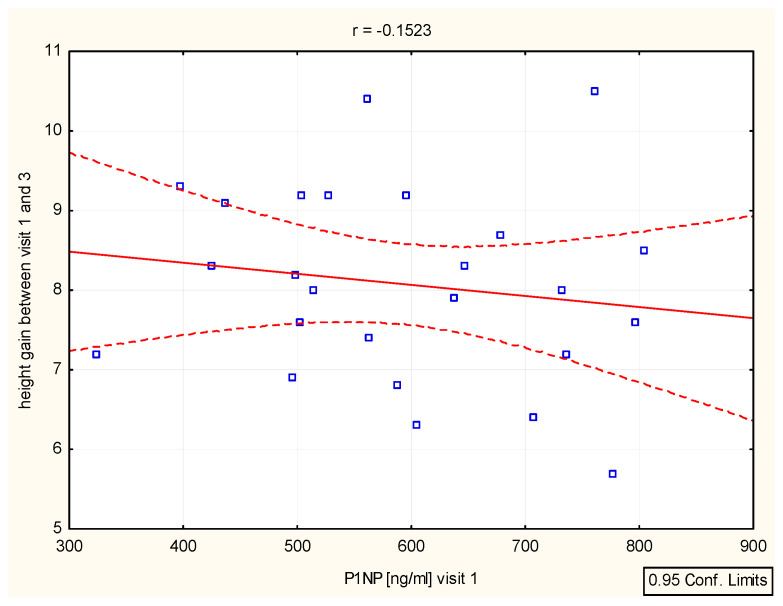
Correlation height/P1NP.

**Table 1 biomedicines-12-01919-t001:** Patients’ characteristics.

	SGA
Age (years)	7.6 ± 1.8
Hbd (weeks)	36.4 Hbd ± 4.11
Born weight (g)	1776.5 ± 699 (−3.3 SD)
Born height (cm)	44.9 ± 6.9 (−1.2 SD)
Height at v1 (cm)	111.8 ± 10.3 (−2.7 SD)
Weight at v1 (g)	16.6 kg ± 3.4 (−3.34 SD)
BMI at v1 (m^2^/kg)	13.1 ± 0.8 (−2 SD)
BA (years)	5.4 ± 1.9

**Table 2 biomedicines-12-01919-t002:** Height, weight, BMI, IGF1, CTX, P1NP, P3NP, and NT-proCNP.

	Height (cm) zscore	Weight (kg) zscore	BMI (m^2^/kg) zscore	IGF1 (ng/mL) zscore	CTX (ng/mL) zscore	P1NP (ng/mL) zscore	P3NP (µg/L) zscore	NT-proCNP (pmol/L) zscore
Visit 1	111.8 (−2.7 SD)	16.68 (−3.34 SD)	13.18 (−2.04 SD)	141	1.7	592	13	57
Visit 2	115.82 (−2.46 SD)	17.99 (−3.06 SD)	13.24 (−2.04 SD)	310	2.4	840	14.1	60
Visit 3	119.93 (−2.19 SD)	19.74 (−2.69 SD)	13.52 (−1.87SD)	349	2.3	804	14.8	58.16
*p* (v1–v3)	*p* = 0.000	*p* = 0.000	*p* = 0.000	*p* = 0.000	*p* = 0.000	*p* = 0.000	*p* = 0.000	n.s.

**Table 3 biomedicines-12-01919-t003:** Calcium and phosphate values.

	Ca (ng/mL)	P (ng/mL)	ALP (ng/mL)	PTH	25OHD3
Visit 1	2.42 ± 0.1	1.49 ± 0.13	231.62 ± 42	22.00 ± 9.11	31 ± 7.7
Visit 2	2.46 ± 0.1	1.62 ± 0.16	277.32 ± 56	29.48 ± 19.22	32.9 ± 7.4
Visit 3	2.46 ± 0.1	1.6 ± 0.14	282.96 ± 52	24.87 ± 17.20	35.18 ± 10
*p* (v1–v3)	0.098	0.001	0.0012	0.03	0.06

**Table 4 biomedicines-12-01919-t004:** Correlation levels and bone markers for CTX, P1NP, and calcium.

	CTX v1	CTX v2	CTX v3	P1NP v1	P1NP v2	P1NP v3	Ca v1	Ca v2	Ca v3
HT cm SD Visit 1	r = 0.1 r = −0.17	r = 0.4 (*p* = 0.047) r = 0.14	r = 0.35 r = −0.08	r = 0.25 r = −0.05	r = 0.03 r = 0.00	r = 0.37 r = 0.38	r = −0.14 r = −0.16	r = −0.10 r = 0.14	r = 0.3 r = 0.53
HT cm SD Visit 2	r = 0.07 r = −0.25	r = 0.39 (*p* = 0.046) r = 0.15 (*p* = 0.049)	r = 0.34 r = −0.10	r = 0.22 r = −0.13	r = 0.03 r = 0.03	r = 0.37 r = 0.41 (*p* = 0.041)	r = −0.13 r = −0.13	r = −0.15 r = 0.16	r = 0.25 0.49
Ht cm SD Visit 3	r = 0.05 r = −0.27	r = 0.4 (*p* = 0.046) r = 0.17	r = 0.33 r = −0.09	r = 0.22 r = −0.10	r = 0.04 r = 0.09	r = 0.38 r = 0.44 (*p* = 0.025)	r = −0.11 r = −0.08	r = −0.20 r = 0.09	r = 0.24 r = 0.48 (*p* = 0.014)

## Data Availability

Data are contained within the article.

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
