# Peer review of "Bone Turnover Markers during Growth Hormone Therapy for Short Stature Children Born Small for Gestational Age"

_biomedicines, 2024, doi:10.3390/biomedicines12081919_

Round 1
Reviewer 1 Report
Comments and Suggestions for Authors
This manuscript reports on growth hormone therapy (daily: subcutaneous administration) in 25 children with short stature SGA (mean age 7.6 years) without any specific disease diagnosis. The authors report that changes in several bone metabolism markers correlated with growth efficiency during the study period (after 0-6-12 months). The quality of the manuscript as a submitted manuscript is somewhat problematic, as the manuscript is poorly written and the exact dosing periods are not stated. The manuscript should be submitted after adequate proofreading.
Major Points:
1. The novelty of the study is not clearly explained in the introduction and following sections.
2. Causative relationships between the bone turnover markers and growth in height are not reasonably explained in the text.
Minor Points:
1. There are several mistypo in the text such as “pepubertal” in line 197, page 7, and “ncreaseed” in line 199, page 7.
2. Abbreviations are not explained appropriately. For example, the explanation of AGA is given much later after it was appeared on the text first time.
Author Response
Dear
Thank You so much for the reviews
1. MAJOR POINTS
1.The novelty of the study is not clearly explained in the introduction and following sections.
The study evaluated the capacity of these markers to predict early growth response to GHT in short stature polish children born with SGA for the first time.
THAT'S THE NOVELTY OF THE STUDY, THAT IS FOR THE FIRST TIME BONE TURNOVER MARKERS WERE USED TO ASSED THE EFFECT OF GROWTH HORMONE THERAPY IN POLISH SGA CHILDREN.
2.Causative relationships between the bone turnover markers and growth in height are not reasonably explained in the text.
I DO NOT UNDERSTAND THIS COMMENTS, BECOUSE THE BONE TURNOVER MARKERS CAN BE USED (AS IGF1 FOR EXEMPL. ) AS MARKERS OF THE EFFECT OF GROWTH HORMONE.
2.MINOR POINTS
1 There are several mistypo in the text such as “pepubertal” in line 197, page 7, and “ncreaseed” in line 199, page 7.
I REALLY DO NOT SEE THIS
That is page 7 in manuscript
P1NP markers and growth velocity.
In Shalender study, early serum P3NP changes were associated with GH administration and could be a useful early predictive biomarker of anabolic response to GH, but the study was not conducted in SGA children [29]. In our study P3NP marker increased after one year of our GH therapy.
NT-proCNP was estimated as a strong marker correlated with growth velocity in children. Prickett in his study showed a direct association of NT-proCNP with the growth process. Plasma NT-proCNP was elevated at birth and fell progressively with age. The authors also emphasised the significant association between NT-proCNP and ALP level as well as Ctx [30]. Unfortunately, we did not find any important increase in NT-proCNP or any correlation with growth velocity or IGF1.
As for the Ca-P metabolism, we found statistically significant elevation of phosphorus and ALP concentrations; Ca levels were positively correlated with growth after one year of treatment.
Bone markers are very poorly documented in children, especially in SGA patients. Even if GH and IGF1 are major factors influencing their levels, other factors are probably also involved.
The study limitations apply to the number of patients and perhaps the length of therapy could also be a limitation of the research.
5.Conclusion
A strong reaction of bone resorption and bone formation markers during growth hormone therapy may determine their selection as prediction of this treatment outcome in SGA children, However, the issue requires further research. 2.Abbreviations are not explained appropriately. For example, the explanation of AGA is given much later after it was appeared on the text first time. I CORRECTEDReviewer 2 Report
Comments and Suggestions for Authors
Review for biomedicines-2954949
Authors: Alicja et al.
In this review article entitled “Bone Turnover Markers during Growth Hormone Therapy Short Stature Children Born as SGA”, the authors evaluated the capacity of bone turnover markers to predict early growth response to GHT in short stature children born with SGA. Furthermore, they checked the implication of Ca-P metabolism.
Hereafter some comments revealed after reviewing the manuscript.
1- Complete titles should be given for both tables 3 and 4, not only “Ca-P” and “Correlations”, respectively.
2- In the references list, each single reference number is duplicated.
3- Some figures, such as fig. 1-3, have very small size of annotations/labels. Please increased the font size.
4- The authors can support the first sentence of introduction “Bone turnover markers…in osteoporosis.” With the following relevant references Doi: 10.1016/j.ejps.2017.04.023 and Doi: 10.1016/j.tox.2020.152412
5- The limitation of the study is overlooked. It would be better to add the limitations of the study. This would have additional value.
6- English language is acceptable just small editing is required.
Minor comments:
1- Use capital letter for liter (“L”) in the whole manuscript including its illustrations, such as “IGF1 (ng/ml)” and “(ng/ml)” in tables 2 and 3, respectively.
2- Check the unit of BMI at V1 and BMI in tables 1 and 2. It should be “m2/ kg”. Use superscript.
Comments on the Quality of English LanguageEnglish language is acceptable just small editing is required.
Author Response
Dear Thank You so much for the reviews The manuscript was corrected in accordance with the review: 1.Titles for both tables was completed ( 3 and 4). 2. All the references list, each single reference number was single ( not duplicated) and stay like that. 3.The figures 1-3 were increased. 4.The sentence of introduction was corected (“Bone turnover markers...in osteoporosis”) 5.The limitation of the study was added. 6. I do not know which specific sentences should be corrected according to English ? Minor comments: 1.“L” was corrected. 2.BMI was from the beginning “m2/kg” , so it is.Round 2
Reviewer 1 Report
Comments and Suggestions for Authors
This study reports on the relationship between the therapeutic effect of GHT on 25 children with SGA and biochemical bone metabolism marker values. However, several issues have to be proven to clarify the manuscript.
Major Points
1. The novelty of this manuscript, which examines the relationship between the effect of treatment with GHT on 25 children with SGA and changes in biochemical bone metabolism markers, is that it was conducted on Polish subjects.
2. The positive finding is that the degree of height gain was inversely correlated with CTX and P1NP levels before treatment. Unfortunately, there are no mechanistic comments about this finding, as the somatomedin induced by GHT stimulated the proliferation and differentiation of osteoblasts that leads to activation of bone formation. The beneficial effect of GHT may have been greater in patients with lower osteogenic activity before treatment, for whom GHT can influence the activity of growing osteoblasts in patients. However, there is no such statement in the text.
3. The GHT performed in this study was initiated in 7-8 year old children. Although uniformity in the timing of treatment initiation seems to be an important point, it is not stated in the text on what basis this was decided.
Minor Issues
1. the typo error noted in the last issue has not been corrected.
The third paragraph of Discussion: “A few studies estimate the improvement of bone strength during GH therapy in SGA children. Willemsen study showed that during long-term GH treatment in short SGA children bone mineral density increased. That was most prominent in children who started treatment at a younger age and in those whith greater height gain during GH treatment [22]. The same concluison we can find in de Zegher study, the rate of bone maturation in short, pepubertal children born SGA treated with GH appeared to depend not only on the dose of GH but also on the age of the child [23]. Lem proved that during GH treatment in SGA children bone mineral density (BMD) ncreased significantly [31]. The same conclusion had the Arends study, when GH treatment in short children born SGA increased proportionately to the height gain, bone maturation[32].”
2. Ca-P levels are discussed, but since the cases were selected after excluding bone metabolic diseases, I do not understand the reason for considering them. There is no association with treatment efficacy, so there seems to be no point in including it in this manuscript.
3. Insufficient description of results and explanation of figures and tables (Legend).
Author Response
Thank you very much for your valuable comments and insightful assessment.Major Points
1, 2 i 3: I introduced these very valuable comments into the text (they are underlined). Many thank's for that.
Minor Points
1.I corrected the third paragraph, hope appropriately. 2.I agree that Ca P is controversial, but I left it as an additional issue.I hope it won't be very controversial. 3.I added the legend. Once again thank You so much.
Reviewer 2 Report
Comments and Suggestions for Authors
I think the manuscript has been improved following the reviewers’ comments. Still 2 minor modifications are needed before going further.
1. In the giving title for table 4, It would be better to use “level” instead of “height” for “Correlations height and…”.
2. Check the unit of BMI at V1 and BMI in tables 1 and 2, respectively. It should be “m2/ kg”. Use superscript. This minor comment for the original version was ignored.
Comments on the Quality of English LanguageThe English language is fine just small editing is required.
Author Response
Thank you very much for your valuable comments 1.I corrected the title of table no. 4, the “level” was added (it is underlined). 2. BMI unit at V1 and in tables 1 and 2 is in m2/kg (it is underlined). Many thank's again.
Round 3
Reviewer 1 Report
Comments and Suggestions for Authors
In revising the version of this manuscript, the authors declared that the novelty of this MS is limited only to the point that it was done on Polish subjects. If the journal accepts this as an original article, the authors should revise the sentences more carefully because many points are ambiguous or unclear, as follows;
Major Points
1. Technical terms must be properly used in the text. For example, the name of biochemical bone turnover markers is normally described as “P1NP”, “CTX”, “ICTP”, “P3NP”, etc. in the abstract. However, in the text, the authors often describe them in many different ways such as “Ctx”, “CTx”, and “PIIINP”. Also, the symbolic name of the gene must be described according to the rule. It is incorrect to write “COLIA1”; it must be written “COL1A1.”
2. The authors describe that P3NP was measured by IRA using the “UniQ PIINP Reagent Kit”. However, it is questionable whether this reagent can correctly detect P3NP. The authors should provide a reference including the company name and catalog number and fully explain that this reagent can detect it.
3. Figure legends are missing.
Minor Points
4. Still, there are a lot of mistypes in the text. The authors should ask the English editing company to correct them.
5. Many abbreviations are used in this manuscript, and the full spelling is not given for some of them. Examples are ECLIA, ELISA, BMI, GHD, ISS, GGTP, IGF, LFP, HCG, ALP, PTH and 25OHD3.
Author Response
Dear,
Many thank's for Your revision.
Major Points;
- I have corrected the bone turnover markers.
- The laboratory methods were also corrected.
- Figure legends as well.
Minor Points;
-
I 've corrected the English (I hope it is well now)
-
All abbreviations were done.